· PLOS | ONE

# Cooperation with autonomous machines through culture and emotion

**Celso M. de Melo** [1]*, **Kazunori Terada** [2]

**1** CCDC U.S. Army Research Laboratory, Playa Vista, CA, United States of America, **2** Gifu University, Gifu, Yanagido, Japan

* celso.miguel.de.melo@gmail.com

**Data Availability Statement:** All data collected and analyzed during this study is available in the SI.

**Funding:** This research was supported by JSPS KAKENHI Grant Number JP16KK0004 [KT], and the US Army. The funder had no role in study

## Abstract

As machines that act autonomously on behalf of others–e.g., robots–become integral to society, it is critical we understand the impact on human decision-making. Here we show that people readily engage in social categorization distinguishing humans ("us") from machines ("them"), which leads to reduced cooperation with machines. However, we show that a simple cultural cue–the ethnicity of the machine's virtual face–mitigated this bias for participants from two distinct cultures (Japan and United States). We further show that situational cues of affiliative intent–namely, expressions of emotion–overrode expectations of coalition alliances from social categories: When machines were from a different culture, participants showed the usual bias when competitive emotion was shown (e.g., joy following exploitation); in contrast, participants cooperated just as much with humans as machines that expressed cooperative emotion (e.g., joy following cooperation). These findings reveal a path for increasing cooperation in society through autonomous machines.

## Introduction

Humans often categorize others as belonging to distinct social groups, distinguishing "us" versus "them", and this categorization influences cooperation, with decisions tending to favor in-group members and, at times, discriminating against out-group members [1–4]. As autonomous machines–such as self-driving cars, drones, and robots–become pervasive in society [5–7], it is important we understand whether humans also apply social categories when engaging with these machines, if decision making is shaped by these categories and, if so, how to overcome unfavorable biases to promote cooperation between humans and machines. Here we show that, when deciding whether to cooperate with a machine, people engage, by default, in social categorization that is unfavorable to machines but, it is possible to override this bias by having machines communicate cues of affiliative intent. In our experiment, participants from two distinct cultures (Japan and United States), engaged in a social dilemma with humans or machines that had a virtual face from the same culture or not and, additionally, expressed emotion conveying cooperative or competitive intent. The results confirmed that people cooperated less with machines than with humans perceived to be from a different culture, except when the emotion indicated an intention to cooperate. Our findings strengthen earlier research indicating that humans rely on social categories–such as culture–to detect coalitional

design, data collection and analysis, decision to
publish, or preparation of the manuscript.

**Competing interests:** The authors have declared
that no competing interests exist.

alliances [8–10] and show that this mechanism applies to autonomous machines. The results
further confirm that it is possible to override these default encodings with situational cues of
affiliative intent [10, 11]–in our case, communicated through emotion. The results also have
important practical implications for the design of autonomous machines, indicating that it is
critical to understand the social context these machines will be immersed in and, adopt mecha-
nisms to convey affiliative intent in order to minimize unfavorable biases and promote cooper-
ation with humans.

In social interaction, people categorize others into groups while associating, or self-identify-
ing, more with some–the in-groups–than others–the out-groups [1–4]. This distinction
between "us" and "them" can lead to a bias that favors cooperation with in-group members [2,
4]. An evolutionary justification for such a bias is to promote prosperity of the in-group which,
in turn, leads to increased chance of survival and longer-term benefits for the individual [12].
In fact, perceptions of group membership have consistently been found to be effective in pro-
moting cooperation in social dilemmas [13]. But, do humans engage in social categorization
when engaging with autonomous machines?

Experimental evidence suggests that people categorize machines similarly to how they do
with other people: in one experiment, in line with gender stereotypes, people assigned more
competence to computers with a female voice than a male voice on the topic of "love and rela-
tionships" [14]; in another experiment, people perceived computers with a virtual face of the
same race as being more trustworthy and giving better advice than a computer with a face of a
different race [15]; in a third experiment, machines with voices that had an accent of the same
culture or not as the participants, impacted perceptions of the appropriateness of the
machine's decisions in social dilemmas [16]. Findings such as these led Reeves and Nass [17]
to propose a general theory arguing that to the extent that machines display human-like cues
(e.g., human appearance, verbal and nonverbal behavior), people will treat them in a funda-
mentally social manner and automatically apply the same rules they use when interacting with
other people. A strict interpretation of this theory would, thus, suggest that not only can people
apply categories to machines, but machines are in-group members.

However, studies show that, despite being able to treat machines as social actors, people
make different decisions and show different patterns of brain activation when engaging with
machines, when compared to humans. As detailed in our recent review of this research [18]:
"Gallagher et al. [19] showed that when people played the rock-paper-scissors game with a
human there was activation of the medial prefrontal cortex, a region of the brain that had pre-
viously been implicated in mentalizing (i.e., inferring of other's beliefs, desires and intentions);
however, no such activation occurred when people engaged with a machine that followed a
known predefined algorithm. McCabe et al. [20] found a similar pattern when people played
the trust game with humans vs. computers, and Kircher et al. [21], Krach et al. [22], and Rilling
et al. [23] replicated this finding in the prisoner's dilemma. (. . .) Sanfey et al. [24] further
showed that, when receiving unfair offers in the ultimatum game, people showed stronger acti-
vation of the bilateral anterior insula–a region associated with the experience of negative emo-
tions–when engaging with humans, when compared to machines." The evidence, thus,
suggests that people experienced less emotion and spent less effort inferring mental states with
machines than with humans. These findings align with research that shows that people per-
ceive less mind in machines than in humans [25]. Denying mind to others or perceiving infe-
rior mental ability in others, in turn, is known to lead to discrimination [26]. Overall, these
findings suggest that machines are treated, at least by default, as members of an out-group.
Effectively, recent studies showed that participants favored humans to computers in several
economic games, including the ultimatum, dictator, and public goods social dilemma [18, 27].
As autonomous machines become pervasive in society, it is important we find solutions to

promote cooperation between humans and machines, including overcoming these types of unfavorable biases.

To accomplish this, we first look at cross categorization, i.e., the idea of associating a positive category with an entity to mitigate the impact of a negative category [3]. Research indicates that humans have the cognitive capacity to process multiple categories simultaneously and crossing categories can reduce intergroup bias [3]. Here we look at culture–pertaining to the shared institutions, social norms, and values of a group of people [9]–as a possible moderator to this bias with machines. Culture is an appropriate first choice in the study of interaction with machines as research indicates that people respond to cultural cues in machines, such as language style [28], accent [16], social norms [29], and race [15]. Research also shows that individuals from different cultures can have different initial expectations about whether the interaction is cooperative or competitive, follow different standards of fairness, and resort to different schemas when engaging in social decision making [30, 31]. Finally, culture has been argued to be important in explaining cooperation among non-kin [8, 32]. Our first hypothesis, therefore, was that associating positive cues of cultural membership could mitigate the default unfavorable bias people have towards machines.

However, it may not always be possible to control the social categories people associate with machines and, thus, it is important to consider a more reliable solution to overcoming negative biases with autonomous machines. Research indicates that, even though social categorization is pervasive, it is possible to override initial expectations of coalitional alliances based on social categories by resorting to more situationally-relevant cues of affiliate intent [10, 11]. Kurzban, Tooby and Cosmides [10] confirmed that people form expectations about coalitions from race but, these were easily overridden by counterparts' verbal statements about intentions to cooperate. They argue social categories are useful only insofar as they are relevant in identifying coalitions. In that sense, cues specific to the social situation should supersede the influence of social categories in perceptions of coalitional alliances. Here we consider emotion expressions for this important social function.

Emotion expressions influence human decision making [33, 34]. One of the important social functions of emotions is to communicate one's beliefs, desires, and intentions to others [35] and, in that sense, emotion displays can be important in identifying cooperators [36]. de Melo, Carnevale, Read and Gratch [37] showed that people were able to retrieve information about how counterparts were appraising the ongoing interaction in the prisoner's dilemma and, from this information, make inferences about the counterparts' likelihood of cooperating in the future. Moreover, emotion displays simulated by machines have also been shown to influence human behavior in other social settings [38]. Our second hypothesis, thus, was that emotion expressions could override expectations of cooperation based on cultural membership.

We present an experiment where participants engaged in the iterated prisoner's dilemma. In this dilemma, two players make a simultaneous decision to either defect or cooperate. Standard decision theory argues that individuals should always defect because defection is the best response to any decision the counterpart may make: if you believe your counterpart will defect, you should defect as well; if you believe your counterpart is going to cooperate, then you still maximize your payoff by defecting. However, if both players follow this reasoning, then they will both be worse off than if they had cooperated. Participants engaged in 20 rounds of this dilemma. Repeating the dilemma a finite number of rounds does not change this prediction since the last round is effectively a one-shot prisoner's dilemma and, by induction, so is every previous round. However, in practice, people often cooperate in such social dilemmas [13]. The payoff matrix we used is shown in Fig 1A. The points earned in the task had real financial consequences as they would be converted to tickets for a $30 lottery. Finally, to prevent any

**A** Payoff Matrix

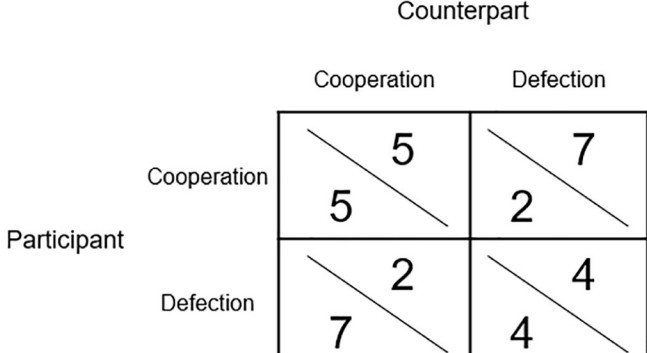

**B** Counterpart Culture & Emotion Expressions

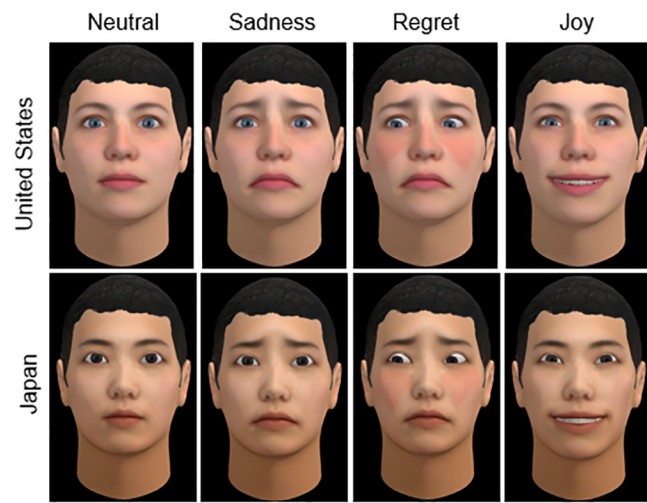

**C** Cooperation Rates

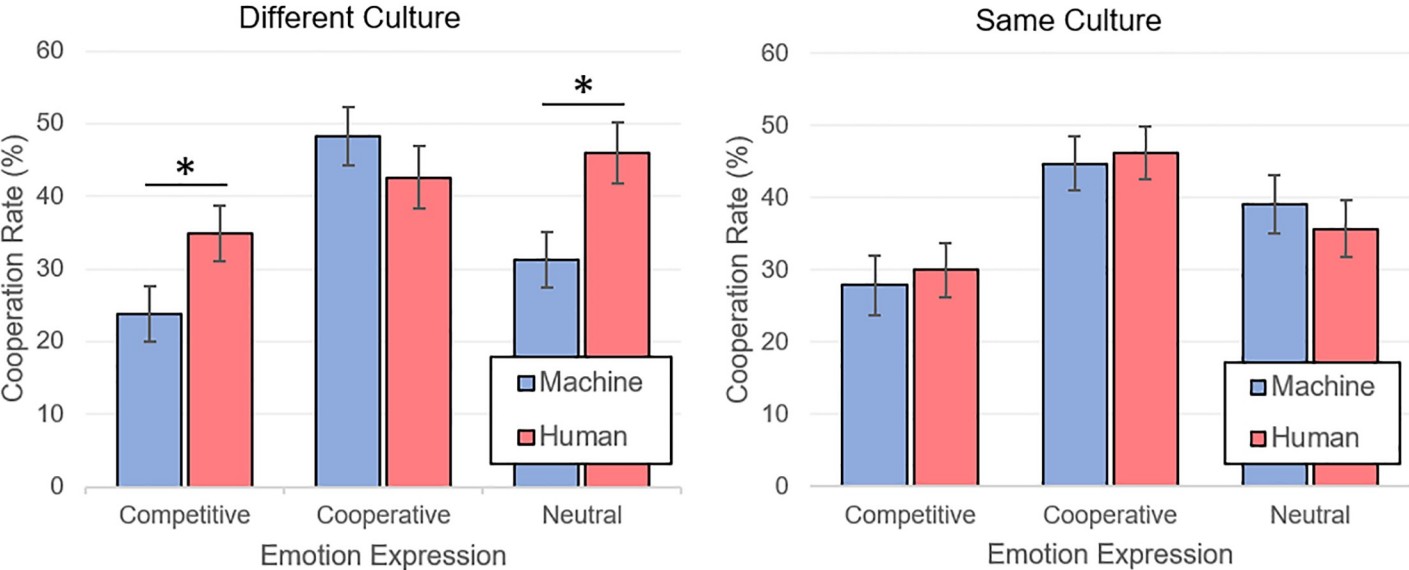

**Fig 1. Experimental manipulations and cooperation rates.** (**A**) The payoff matrix for the prisoner's dilemma, (**B**) Counterparts' virtual faces typical in the United States (top) and Japan (bottom) and corresponding emotion expressions, (**C**) Cooperation rates when the counterpart was from a different culture as the participant (left) or the same culture (right). The error bars correspond to standard errors. * p < .05.

reputation concerns, the experiment was fully anonymous–i.e., the participants were anonymous to each other and to the experimenters (see the Materials and methods section for details on how this was accomplished).

Participants were told they would engage in the prisoner's dilemma with either another participant or with an autonomous machine. In reality, to maximize experimental control, they always engaged with a computer script. Similar methods have been followed in previous studies of human behavior with machines [18, 19, 21, 23, 24] and the experimental procedures were fully approved by the Gifu University IRB. This script followed a tit-for-tat strategy (starting with a defection) and showed a pre-defined pattern for emotion expression (see below). The focus of the experiment was in studying whether participants would cooperate distinctly

with humans vs. machines and, if so, whether culture or emotion expressions could moderate this effect.

To manipulate perceptions of cultural membership, counterparts were given virtual faces that were either typical in the United States or in Japan–see Fig 1B. See the Supporting Information (S1 File) appendix for a validation study, with a separate sample of participants, for perceptions of the corresponding ethnicities in these faces (S1 File). We recruited 945 participants from the United States (*n* = 468) and Japan (*n* = 477) using online pools (see Materials and methods for more details about recruitment and sample demographics). Participants were either matched with a counterpart of the same or different culture, counterbalanced across participants. This manipulation allowed us to study, in two distinct cultures, how participants behaved with (human or machine) counterparts that were either in- or out-group members to the culture.

Counterparts expressed emotion through their virtual faces corresponding to either a competitive, neutral, or cooperative orientation. Building on earlier work that shows that emotion expressions can shape cooperation in the prisoner's dilemma [37], we chose the following patterns: competitive emotions–regret following mutual cooperation (given that it missed the opportunity to exploit the participant), joy following exploitation (participant cooperates, counterpart defects), sadness in mutual defection and, neutral otherwise; cooperative emotions–joy following mutual cooperation, regret following exploitation, sadness in mutual defection, and neutral otherwise; neutral emotions–neutral expression for all outcomes. For a validation study for perception of the intended emotions, please see the SI appendix (S1 File). In sum, we ran a 2 × 2 × 3 between-participants factorial design: *counterpart type* (human vs. machine) × *counterpart culture* (United States vs. Japan) × *emotion* (competitive vs. neutral vs. cooperative). Our main measure was cooperation rate, averaged across all rounds.

## Results

The focus of our analysis was two-fold: (1) understand the cooperation rate when participants engaged with humans vs. machines that had the same vs. different culture; and, (2) understand the moderating role of emotion expressions. To accomplish this, we first split the data into two sets: the first corresponding to pairings of participants with counterparts of a different culture, and the second corresponding to pairings with counterparts of the same culture. For each set, we ran a participant sample (United States vs. Japan) × counterpart type (human vs. machine) × emotion (competitive vs. neutral vs. cooperative) between-participants factorial ANOVA. Fig 1C shows the cooperation rates for this analysis.

When participants were paired with counterparts of a different culture, we found the expected main effect of counterpart type, $F(1, 436) = 4.17$, $P = 0.042$, partial $\eta^2 = 0.01$: participants cooperated more with humans (*M* = 41.15, *SE* = 2.39) than machines (*M* = 34.45, *SE* = 2.25). There was also a main effect of emotion–$F(2, 436) = 8.17$, $P < 0.001$, partial $\eta^2 = 0.04$ –and Bonferroni post-hoc tests showed that: participants cooperated more with cooperative (*M* = 45.42, *SE* = 2.97) than competitive counterparts (*M* = 29.34, *SE* = 2.70), $P < .001$; and, participants tended to cooperate more with neutral (*M* = 38.63, *SE* = 2.85) than competitive counterparts, $P = 0.055$. Interestingly, however, there was a counterpart type × emotion interaction, $F(2, 436) = 3.48$, $P = 0.032$, partial $\eta^2 = 0.16$. To get insight into this interaction, we split the data by emotion condition and ran a participant sample × counterpart type between-participants factorial ANOVA. This analysis revealed that: when counterparts showed competitive or neutral emotions, there was a main effect of counterpart type–respectively, $F(3, 159) = 5.39$, $P = 0.022$, partial $\eta^2 = 0.03$, and $F(3, 146) = 5.83$, $P = 0.017$, partial $\eta^2 = 0.04$ –with participants cooperating more with humans than machines; but, when the counterparts showed

cooperative emotion, there was no statistically significant difference in cooperation between humans and machines, $F(3, 131) = 0.83$, $P = 0.365$. Finally, there was no main effect of participant sample–$F(1, 436) = 1.28$, $P = 0.259$ –and no statistically significant interactions with the other factors, suggesting that the effects apply both for participants in Japan and the United States.

When participants engaged with counterparts of the same culture, in contrast to the previous case, there was no statistically significant main effect of counterpart type–$F(1, 485) = 0.01$, $P = 0.972$ –or counterpart type × emotion interaction–$F(2, 485) = 0.29$, $P = 0.749$. The main effect of emotion, on the other hand, was still statistically significant–$F(2, 485) = 558.71$, $P < 0.001$, partial $\eta^2 = 0.54$ –and Bonferroni post-hoc tests revealed that participants: cooperated more with cooperative ($M = 45.55$, $SE = 2.61$) than competitive counterparts ($M = 28.99$, $SE = 2.78$), $P < .001$; tended to cooperate more with cooperative than neutral counterparts ($M = 37.42$, $SE = 2.81$), $P = 0.104$; and, tended to cooperate more with neutral than competitive counterparts, $P = 0.100$. Once again, there was no main effect of participant sample–$F(1, 485) = 0.97$, $P = 0.813$ –and no statistically significant interactions.

For further insight and analyses, please refer to the SI for a table with all descriptive statistics (S1 Table) and the raw data (S2 File).

## Discussion

Despite the changes autonomous machines promise to bring to society, here we show that humans will resort to familiar psychological mechanisms to identify alliances and collaborate with machines. Whereas autonomous machines may be perceived by default as out-group members [18–27], our experimental results with participants from two distinct cultures (Japan and United States) indicate that simple cues of cultural in-group membership–based on the ethnicity of the machine's virtual face–can mitigate this unfavorable bias in the decisions people make with machines, when compared to humans. More fundamentally, the results indicate that situational cues of affiliative intent–in our experiment, through expressions of emotion–can override default expectations created from social categorization and promote cooperation between humans and machines.

Our results confirm that, in the context of interaction with autonomous machines, social categorization occurs naturally and influences human decision making. Participants cooperated more with machines that were perceived to belong to the same culture than those that were not. Culture has been argued to be central to cooperation with non-kin [8, 32] and our results indicate that this can extend to interactions with machines. This is also in line with earlier research showing that humans readily apply social rules, including social categorization, when interacting with machines in social settings [14–17]. However, the results further show that, when counterparts were perceived to belong to a different culture, participants cooperated less with machines than with humans. This reinforces that, despite being able to treat machines in a social manner, by default, people still show an unfavorable bias with machines, when compared to humans [18–27]; in other words, machines are perceived, by default, as belonging to an out-group. By crossing this default negative category with a positive cue for culture membership, as we do in our experiment, we were able to mitigate this bias, suggesting that multiple social categorization [3] can be a solution for reducing intergroup bias with machines.

A more reliable solution, however, may be to communicate affiliative intent through emotion expression. Emotion had the strongest effect in our experiment, showing that even a machine from a different culture group could be treated like an in-group member through judicious expression of emotion–in our case, joy following cooperation and regret after

exploitation. Emotion expressions have been argued to serve important social functions [35] and help regulate decision making [34, 36, 37], and here we strengthen research indicating that emotion in autonomous machines is a powerful influencer of human behavior [38], including social decision making [27, 37]. More generally, in line with earlier research [10, 11], this research indicates that default coalition expectations from social categories can be overridden if situational coalition information is available. This is encouraging as it may not always be possible to control the social categories that will be perceived in machines.

The work presented here has some limitations that introduce opportunities for future work. First, the effects of emotion expressions, when compared to the neutral (control) condition, could be strengthened. For instance, whereas people cooperated more with cooperative than competitive counterparts, there was only a trend for higher cooperation with cooperative than neutral counterparts. To strengthen these effects, people may need to be exposed longer to the expressions (e.g., by increasing the number of rounds), or an even clearer emotional signal may need to be communicated (e.g., verbal or multimodal expression of emotion [37]). Second, social discrimination among humans is complex and here we have only begun studying this topic. Dovidio and Gartner [39] point out that, rather than blatant and open, modern racism tends to be subtle. In fact, in some cases, to avoid being perceived as racist, people can over-compensate when interacting with out-group members [40]. This may explain why, in our case, participants tended to cooperate more, when no emotion was shown, with humans of a different than the same culture. This compensation mechanism, however, did not occur with machine counterparts, where we see the expected in-group bias–thus, exemplifying that machines are often treated less favorably than humans. It is important, therefore, to understand whether interaction with machines will also evolve to reflect subtler forms of discrimination, especially as they become more pervasive in society. Finally, here we looked at race as a signal for cultural membership, but several other indicators have been studied [15, 16, 28, 29]. Follow-up work should study the relative effects of these different indicators–e.g., speech accent or country of origin–on cooperation with machines. As noted in the SI, perception of ethnicity from race can still lead to some ambiguity, with Japanese participants more easily identifying the Caucasian face as being from the United States than American participants. Multiple complementary signals could, thus, potentially be used to strengthen perceptions of in-group cultural membership and, consequently, help further reduce bias.

The results presented in this paper have practical importance for the design of autonomous machines. The existence of a default unfavorable bias towards machines means designers should take action if they hope to achieve the levels of cooperation seen among humans. It wouldn't be satisfactory to conceal that a machine is autonomous–i.e., is not being directly controlled by a human–as there is increased expectation in society of transparency and interpretability from algorithms [41]. Instead, designers should consider the broader social context and the cognitive mechanisms driving humans to promote cooperation with machines. Here we show that social categorization is pervasive and can be leveraged–through simple visual [15], verbal [16, 28], or behavioral [29] cues–to increase perceptions of group membership with machines and, subsequently, encourage more favorable decisions. However, on the one hand, it may not always be possible to provide those cues and, on the other, social categories can be activated by unexpected cues. In this sense, designers should consider specific situational cues as a more explicit signal of the machine's affiliative intent. Here we exemplify how this signaling can be achieved, effectively and naturally, through expressions of emotion.

At a time of increasing divisiveness in society, it may seem unsurprising that autonomous machines are perceived as being outsiders and, consequently, being less likely to benefit from the advantages afforded to in-group members. However, autonomous machines presumably act on behalf of (one or several) humans who, logically, are the ultimate targets of any decision

made with these machines. Nevertheless, it is encouraging to learn that behavior with these machines appears to be driven by the same psychological mechanisms in human-human interaction. This introduces familiar opportunities for reducing intergroup bias, such as cultural membership priming and cross categorization. It is also comforting to learn that if a machine intends to communicate with humans, and is able to effectively communicate that affiliative intent, then any default expectations derived from social categorization can be overridden. Since autonomous machines can be designed to take advantage of these cognitive-psychological mechanisms driving human behavior, they introduce a unique opportunity to promote a more cooperative society.

## Materials and methods

This section describes details for the experimental methods that are not described in the main body of the text.

### Experimental task

Building on previous work [37], the prisoner's dilemma game was recast as an investment game and described as follows to the participants: "You are going to play a two-player investment game. You can invest in one of two projects: project green and project blue. However, how many points you get is contingent on which project the other player invests in. So, if you both invest in project green, then each gets 5 points. If you choose project green but the other player chooses project blue, then you get 2 and the other player gets 7 points. If, on the other hand, you choose project blue and the other player chooses project green, then you get 7 and the other player gets 2 points. A fourth possibility is that you both choose project blue, in which case both get 4 points". Thus, choosing project green corresponded to the cooperative choice, and project blue to defection. Screenshots of the software are shown in the Supporting Information (S3 and S4 Figs). The software was presented in English to US participants and translated to Japanese for participants in Japan.

### Participant samples

All participants were recruited from online pools: the US sample was collected from Amazon Mechanical Turk, and the Japanese sample from Yahoo! Japan Crowdsourcing. Previous research shows that studies performed in online platforms can yield high-quality data and successfully replicate the results of behavioral studies performed on traditional pools [42]. To estimate sample size per country, we used G*Power 3. Based on earlier work [18, 37], we predicted a small to medium effect size (Cohen's $f$ = 0.20). Thus, for $\alpha$ = .05 and statistical power of .85, the recommended total sample size was 462 participants. In practice, we recruited 468 participants in the US and 477 in Japan. The demographics for the US sample were as follows: 63.2% were males; age distribution– 18 to 21 years, 0.9%; 22 to 34 years, 58.5%; 35 to 44 years, 21.6%; 45 to 54 years, 10.7%; 55 to 64 years, 6.2%; over 64 years, 2.1%; ethnicity distribution–Caucasian, 77.3%; African American, 10.3%; East Indian, 1.3%; Hispanic or Latino, 9.2%; Southeast Asian, 6.0%. The demographics for the Japanese sample were as follows: 67.4% were males; age distribution– 18 to 21 years, 0.6%; 22 to 34 years, 15.7%; 35 to 44 years, 36.4%; 45 to 54 years, 34.7%; 55 to 64 years, 10.4%; over 64 years, 2.1%; ethnicity distribution–East Indian, 0.6%; Southeast Asian, 99.4%.

### Financial incentives

Participants in the US were paid $2.00 for participating in the experiment, whereas participants in Japan were paid 220 JPY (~$2.00). Moreover, they had the opportunity to earn more

money according to their performance in the task. Each point earned in the task was converted to a ticket for a lottery worth $30.00 for the US sample and 3,000 JPY (~$27.00) for the Japanese sample.

## Full anonymity

All experiments were fully anonymous for participants. To accomplish this, counterparts had anonymous names and we never collected any information that could identify participants. To preserve anonymity with respect to experimenters, we relied on the anonymity system of the online pools we used. When interacting with participants, researchers are never able to identify the participants, unless we explicitly ask for information that may serve to identify them (e.g., name, email, or photo), which we did not. This experimental procedure is meant to minimize any possible reputation effects, such as a concern for future retaliation for the decisions made in the task.

## Data analyses

The main analysis consisted of participant sample (United States vs. Japan) × counterpart type (human vs. machine) × emotion (competitive vs. neutral vs. cooperative) between-participants factorial ANOVAs on average cooperation rate, for the case where participants were matched with counterparts of the same culture and the case where counterparts had a different culture. To get insight into statistically significant interactions, we ran Bonferroni post-hoc tests and follow-up participant sample × counterpart type between-participants factorial ANOVAs, where the data was split on emotion condition.

## Ethics

All experimental methods were approved by the Medical Review Board of Gifu University Graduate School of Medicine (IRB ID#2018–159). As recommended by the IRB, written informed consent was provided by choosing one of two options in the online form: 1) "I am indicating that I have read the information in the instructions for participating in this research and have had a chance to ask any questions I have about the study. I consent to participate in this research.", or 2) "I do not consent to participate in this research." All participants gave informed consent and, at the end, were debriefed about the experimental procedures.

## Supporting information

**S1 Fig. Perception of emotion in Japanese and Caucasian faces.**
(TIF)

**S2 Fig. Perception of ethnicity in Japanese and Caucasian faces by US and Japanese participant samples.**
(TIF)

**S3 Fig. The prisoner's dilemma software for the United States participants.** The counterpart in this case has the same culture and is showing cooperative emotions.
(TIF)

**S4 Fig. The prisoner's dilemma software for the Japanese participants.** The counterpart in this case has the same culture and is showing competitive emotions.
(TIF)

**S1 File. Appendix with validation experiment for emotion and ethnicity perception in virtual faces.**
(DOCX)

**S2 File. CSV file with raw data.**
(CSV)

**S1 Table. Descriptive statistics for main experiment.**
(DOCX)

## Author Contributions

**Conceptualization:** Celso M. de Melo, Kazunori Terada.

**Data curation:** Celso M. de Melo.

**Formal analysis:** Celso M. de Melo, Kazunori Terada.

**Funding acquisition:** Celso M. de Melo, Kazunori Terada.

**Investigation:** Celso M. de Melo, Kazunori Terada.

**Methodology:** Kazunori Terada.

**Resources:** Celso M. de Melo, Kazunori Terada.

**Software:** Celso M. de Melo.

**Validation:** Celso M. de Melo, Kazunori Terada.

**Writing – original draft:** Celso M. de Melo, Kazunori Terada.

**Writing – review & editing:** Celso M. de Melo, Kazunori Terada.

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
