## [Decision Letter · Decision Letter 0]

10 Sep 2019

PONE-D-19-20447

Cooperation with Autonomous Machines Through Culture and Emotion

PLOS ONE

Dear Dr. de Melo,

Thank you for submitting your manuscript to PLOS ONE. After careful consideration by two expert reviewers, we feel that the manuscript has merit but does not fully meet PLOS ONE’s publication criteria as it currently stands. Nonetheless, you will see that both reports are positive, yet suggesting some additional clarifications and modifications before acceptance. Therefore, we invite you to submit a revised version of the manuscript that addresses each of the points raised during the review process.

We would appreciate receiving your revised manuscript by Oct 25 2019 11:59PM. To enhance the reproducibility of your results, we recommend that if applicable you deposit your laboratory protocols in protocols.io, where a protocol can be assigned its own identifier (DOI) such that it can be cited independently in the future. For instructions see: http://journals.plos.org/plosone/s/submission-guidelines#loc-laboratory-protocols

We look forward to receiving your revised manuscript.

Kind regards,

Francisco C. Santos

Academic Editor

PLOS ONE

Journal Requirements:

2. We noticed you have some minor occurrence of overlapping text with your previous publications:

Melo, Celso De, Stacy Marsella, and Jonathan Gratch. "People do not feel guilty about exploiting machines." ACM Transactions on Computer-Human Interaction (TOCHI) 23.2 (2016): 8.

de Melo, Celso M., and Jonathan Gratch. "People show envy, not guilt, when making decisions with machines." 2015 International Conference on Affective Computing and Intelligent Interaction (ACII). IEEE, 2015.

The text that needs to be addressed involves the fourth paragraph of the introduction.

In your revision ensure you cite all your sources (including your own works), and quote or rephrase any duplicated text outside the methods section.

3.  Please provide additional details regarding participant consent.

In the ethics statement in the Methods and online submission information, please ensure that you have specified what type of consent you obtained (for instance, written or verbal, and if verbal, how it was documented and witnessed).

Reviewers' comments:

Reviewer's Responses to Questions

**Comments to the Author**

1. Is the manuscript technically sound, and do the data support the conclusions?

Reviewer #1: Yes

Reviewer #2: Partly

2. Has the statistical analysis been performed appropriately and rigorously? 

Reviewer #1: Yes

Reviewer #2: Yes

3. Have the authors made all data underlying the findings in their manuscript fully available?

Reviewer #1: Yes

Reviewer #2: Yes

4. Is the manuscript presented in an intelligible fashion and written in standard English?

Reviewer #1: Yes

Reviewer #2: Yes

5. Review Comments to the Author

Reviewer #1: This manuscript ("Cooperation with Autonomous Machines Through Culture and Emotion") addresses the subject of how human decision-making is affected by interactions with autonomous machines. Throughout the paper, the authors present several results that are able to indicate that there are significant differences between the levels of cooperation in a Prisoner's Dilemma when the game is played among Humans in contrast to when the counterpart is a machine. These differences seem to be related to similar psychological mechanisms that also occur when humans interact with members of different groups/cultures. Nevertheless, the experimental results show that the presence of cooperative emotional cues are able to overcome these situations and increase the levels of cooperation.

I believe the questions raised in this paper, as well as the findings are both relevant and very important. All statistical analysis and experimental procedures appear to be correct and follow strict scientific methodology. Moreover, this paper contributes to the increasingly important understanding of the interactions of hybrid human-agent societies. Finally, their conclusions over the psychological mechanisms of human-machine interactions offer a range of potential applications, including nudging cooperation in human societies through autonomous machines, an application that is also supported by other experimental research [e.g., (Shirado et al., 2019)]. Therefore, I consider that this manuscript should be accepted for publication.

However, I do have some minor remarks that may improve the readability of the paper as well as some questions about the conclusions:

1. In Page 7 the authors mention: "...The problem is that if both players think like this, then they will both be worse off than if they had both cooperated. "

This explanation may result slightly confusing, I would instead rephrase it as:

"...However if both players follow this reasoning, then they will both be worse off than if they had cooperated."

2. In page 7 you mention: "Participants were told they would engage in the prisoner's dilemma with either another participant or with an autonomous machine. In reality, to maximise experimental control, they always engaged with a computer script that followed a tit-for-tat strategy..."

It is not very clear whether your participants never played against another human or, when engaging against a machine, the machine always used a tit-for-tat strategy. I had to read much further to understand it.

3. In page 8, when you describe competitive emotions, it could be useful for the readers to indicate why regret following mutual defection is part of a competitive emotion. Perhaps because it indicates that the counterpart regrets not defecting after knowing that the participant cooperated?

4. The result section could perhaps perhaps use a bit of rewriting. Sometimes the statistical results are indicated in parenthesis, others in between commas, which difficult the reading flow. However, I don't deem this to be extremely important, and the text as a whole is still sufficiently understandable and correct.

One last question:

In Figure 1C, the cooperation rate between humans of the same and different cultures in a competitive environment does not seem to differ significantly, does this mean that humans treat "equally" (perhaps as members of a different group) all counterparts when in a competitive emotional environment?

Reviewer #2: In this article, the authors report the findings of a cross-cultural study involving the United States and Japan where participants from these two countries interacted online with a virtual agent in an iterated prisoner’s dilemma lasting 20 rounds. The aim of the study was to gain insight on how social categorization as well as emotion expression affects human decision-making. To this effect, the authors ran a between participants study where they manipulated the appearance of the agent’s face as a cultural cue as well as its emotional displays during the interaction. Additionally, the authors also manipulated the perception of autonomy with one group being told they were interacting with an avatar of another human and another group being informed that the agent was acting autonomously. The motivation and methodology of the study draws heavily from other similar work in this line of human-agent interaction research, which uses well established scenarios from game theory to study how humans cooperate with artificial entities. This type of research is important to help us understand how humans and autonomous machines can collaborate with one another. The authors provide a fair number of citations to previous work that helps to situate the novelty of the study presented here. The main novel aspect of the study is that it analyses the interplay between how the agent is socially categorized according to its appearance and the emotional signals it decides to give when playing the game using the Tit-for-tat strategy. Overall the paper is quite well written and easy to follow. Also, from a methodological standpoint, I commend the fact that the authors conducted a power analysis to determine what would be a proper sample size and ended up with more than 400 participants for each country, which greatly increases the robustness of the obtained results. With that said, I do have some criticisms that I would like the authors to address. Firstly, while the results show in fact support for the conclusion that the competitive emotional signalling leads to significantly less cooperation it seems that there is no significant main effect between the cooperative and the neutral strategy for emotional expression. If the goal is to increase cooperation one would not hypothesize that the competitive emotional strategy would be a suitable approach. The obtained results clearly confirm that it is not. However, from a design standpoint and as mentioned by the authors in their motivation, the main research question here is whether having a cooperative emotional strategy can lead to an increased degree of cooperation compared to not showing emotions at all. From that perspective, the results should be discussed more in depth. For instance, when the culture is different, the degree of cooperation in the human condition was roughly the same in both the cooperative emotional strategy and the neutral one. What possible reasons the authors think can explain this result that could be tested in a future study? Perhaps this was due to different cultural expectations of when one should show emotion that are applied more strongly when interacting with other humans. Additionally, in the neutral emotional strategy, the degree of cooperation was higher in the different culture condition than it was in the same culture condition for the human counterpart. This is also an unexpected result according to the in-group hypothesis. Possibly this has to do with using the appearance of the character’s face as a single cultural cue. As reported in the appendix discussing the validation of the ethnicity perception, the authors do report that US participants were significantly less likely to perceive any particular ethnicity in the face than participants from Japan. Perhaps, adding an additional cultural cue, such as having an iconic background image placed behind the character would reinforce the social categorization process. Overall, I think the paper is quite interesting and relevant but a more in-depth discussion of the results as well as a few paragraphs of limitations and future work is greatly warranted. Finally, the authors should also include the effect size when describing their results.

6. PLOS authors have the option to publish the peer review history of their article (what does this mean?). If published, this will include your full peer review and any attached files.

Reviewer #1: No

Reviewer #2: No

---

## [Author Response · Author response to Decision Letter 0]

22 Sep 2019

Dear Editor,

Thank you for the thorough and constructive review of our manuscript “Cooperation with Autonomous Machines Through Culture and Emotion” that we submitted to PLOS ONE for possible publication. We acknowledge and highly appreciate the reviewers’ comments and, to address them, we would like to submit a careful revision of the manuscript, Supporting Information, and other materials. 

In the following, we present a point-by-point summary of how we have addressed each issue raised by the reviewers. Regarding Reviewer 1’s comments:

1. “In Page 7 the authors mention: ‘...The problem is that if both players think like this, then they will both be worse off than if they had both cooperated.’ This explanation may result slightly confusing, I would instead rephrase it as: ‘...However if both players follow this reasoning, then they will both be worse off than if they had cooperated.’”

We changed the text according to the reviewer’s suggestion. 

2. “In page 7 you mention: ‘Participants were told they would engage in the prisoner's dilemma with either another participant or with an autonomous machine. In reality, to maximize experimental control, they always engaged with a computer script that followed a tit-for-tat strategy...’It is not very clear whether your participants never played against another human or, when engaging against a machine, the machine always used a tit-for-tat strategy. I had to read much further to understand it.”

To clarify this, we changed the text as follows: “In reality, to maximize experimental control, they always engaged with a computer script. (…) This script followed a tit-for-tat strategy (starting with a defection) and showed a pre-defined pattern for emotion expression (…)”

3. “In page 8, when you describe competitive emotions, it could be useful for the readers to indicate why regret following mutual defection is part of a competitive emotion. Perhaps because it indicates that the counterpart regrets not defecting after knowing that the participant cooperated?”

This is exactly right and is supported by our prior work (de Melo et al., 2014), which is referred to in the same paragraph. Accordingly, we updated the text as follows: “(…) competitive emotions – regret following mutual cooperation (given that it missed the opportunity to exploit the participant), joy following exploitation (…)”.

4. “The results section could perhaps use a bit of rewriting. Sometimes the statistical results are indicated in parenthesis, others in between commas, which difficult the reading flow.”

We revised the results section to present a consistent format for the statistical results. We also fixed the means and standard errors to match the 0 to 100 range in Figure 1-C. We hope these changes will help the reading flow. 

5. “In Figure 1C, the cooperation rate between humans of the same and different cultures in a competitive environment does not seem to differ significantly, does this mean that humans treat "equally" (perhaps as members of a different group) all counterparts when in a competitive emotional environment?”

Indeed, there is no statistically significant difference in cooperation rate, when competitive emotions are shown, between humans of different (M = 34.46, SE = 3.57) and same culture (M = 30.11, SE = 3.11), t(171) = 0.92, P = 0.169. This result is, nevertheless, in line with the argument presented in the paper that, when there is situational information about (lack of) affiliative intent, this information should supersede the effect of social categorization. However, as this and some of the other reviewer’s comments emphasize, there are several subtleties and possibly interesting effects in the data that are not necessarily the main focus of the paper. In that sense, we added a table to the SI with all descriptive statistics for our results. Moreover, all the data is now available as part of the SI, to support any additional analyses that the readers may wish to perform. (Please also see our reply to related comments made by Reviewer 2.)

Regarding Reviewer 2’s comments: 

1. “Firstly, while the results show in fact support for the conclusion that the competitive emotional signaling leads to significantly less cooperation it seems that there is no significant main effect between the cooperative and the neutral strategy for emotional expression. If the goal is to increase cooperation one would not hypothesize that the competitive emotional strategy would be a suitable approach. The obtained results clearly confirm that it is not. However, from a design standpoint and as mentioned by the authors in their motivation, the main research question here is whether having a cooperative emotional strategy can lead to an increased degree of cooperation compared to not showing emotions at all. From that perspective, the results should be discussed more in depth.” 

We agree that, from a practical perspective, it is important to understand if cooperative emotion displays increase cooperation with respect to the baseline neutral emotion case. In our sample, in line with the reviewer’s comments, we only see trends in that direction: different culture, P = 0.299; and, as reported in the text, same culture, P = 0.104. Failure to get an effect as strong as reported in our earlier work (e.g., de Melo et al., 2014) may have happened because the emotional signal was simply not strong enough. 

Accordingly, we added a paragraph to the discussion that addresses this limitation and proposes future work (pg. 12): “The work presented here has some limitations that introduce opportunities for future work. First, the effects of emotion expressions, when compared to the neutral (control) condition, could be strengthened. For instance, whereas people cooperated more with cooperative than competitive counterparts, there was only a trend for higher cooperation with cooperative than neutral counterparts. To strengthen these effects, people may need to be exposed longer to the expressions (e.g., by increasing the number of rounds), or an even clearer emotional signal may need to be communicated (e.g., verbal or multimodal expression of emotion [37]).”

2. “(…) when the culture is different, the degree of cooperation in the human condition was roughly the same in both the cooperative emotional strategy and the neutral one. What possible reasons the authors think can explain this result that could be tested in a future study? Perhaps this was due to different cultural expectations of when one should show emotion that are applied more strongly when interacting with other humans. Additionally, in the neutral emotional strategy, the degree of cooperation was higher in the different culture condition than it was in the same culture condition for the human counterpart. This is also an unexpected result according to the in-group hypothesis. Possibly this has to do with using the appearance of the character’s face as a single cultural cue. As reported in the appendix discussing the validation of the ethnicity perception, the authors do report that US participants were significantly less likely to perceive any particular ethnicity in the face than participants from Japan. Perhaps, adding an additional cultural cue, such as having an iconic background image placed behind the character would reinforce the social categorization process. Overall, I think the paper is quite interesting and relevant but a more in-depth discussion of the results as well as a few paragraphs of limitations and future work is greatly warranted.” 

These are interesting observations about behavior with humans that we had not focused in the paper. When no emotion is shown, the straightforward implication for the in-group bias is higher cooperation with counterparts of the same culture than different culture. We see that pattern with machines, but we see an opposite trend with humans. Social discrimination is a complex phenomenon and, as pointed out by Dovidio and Gaertner (2000), racism can manifest in rather subtle ways. As further noted by Axt, Ebersole, and Nosek (2016), to avoid being perceived negatively, people can over-compensate when interacting with out-group members. Thus, in the absence of situational information about affiliative intent (i.e., no emotion), people would be more generous with out-group members than they would otherwise be. This being a sophisticated “masking” mechanism, it is unsurprising that it was not engaged with machines, which is consistent with the idea of an unfavorable default bias with machines.

The alternative mechanism of different cultural expectations about the display of emotion is interesting and, though there is research about cultural emotion display rules, we are not aware of research indicating that avoiding expression of emotion can increase cooperation in inter-group decision making. However, this is an interesting line of inquiry.

The second alternative explanation pertained to using the face as the single cue for cultural membership. However, our interpretation of the results is that neutral out-group humans were being treated more favorably than in-group members. In that sense, even though American participants perceived multiple ethnicities in the Caucasian face, what is critical is that they were able to perceive the Japanese face as belonging to an out-group ethnicity. Accordingly, both American and Japanese participants favored neutral out-group humans to neutral in-group humans. Nevertheless, we acknowledge the value of studying alternative and multiple signals for cultural membership in the revised discussion.

Following this comment, we added the following paragraph to the discussion (pgs. 12-13): “(…) social discrimination among humans is complex and here we have only begun studying this topic. Dovidio and Gartner [39] point out that, rather than blatant and open, modern racism tends to be subtle. In fact, in some cases, to avoid being perceived as racist, people can over-compensate when interacting with out-group members [40]. This may explain why, in our case, participants tended to cooperate more, when no emotion was shown, with humans of a different than the same culture. This compensation mechanism, however, did not occur with machine counterparts, where we see the expected in-group bias – thus, exemplifying that machines are often treated less favorably than humans. It is important, therefore, to understand whether interaction with machines will also evolve to reflect subtler forms of discrimination, especially as they become more pervasive in society. Finally, here we looked at race as a signal for cultural membership, but several other indicators have been studied [15, 16, 28, 29]. Follow-up work should study the relative effects of these different indicators – e.g., speech accent or country of origin – on cooperation with machines. As noted in the SI, perception of ethnicity from race can still lead to some ambiguity, with Japanese participants more easily identifying the Caucasian face as being from the United States than American participants. Multiple complementary signals could, thus, potentially be used to strengthen perceptions of in-group cultural membership and, consequently, help further reduce bias.”

Finally, acknowledging comments by both reviewers, we have added a table to the SI with descriptive statistics for the experimental results, as well as shared all data in the SI. We hope this will encourage further independent inquiry of the results presented in the paper.

3. Finally, the authors should also include the effect size when describing their results.” 

The original text was, indeed, missing some of the effect sizes. We fully revised the results section to report the effect size (partial eta squared value) for all main effects and interactions from our ANOVA analyses.

***

Please do not hesitate to contact us if any questions should arise during the review process.

Kind regards,

Celso M. de Melo and Kazunori Terada

---

## [Decision Letter · Decision Letter 1]

22 Oct 2019

Cooperation with Autonomous Machines Through Culture and Emotion

PONE-D-19-20447R1

Dear Dr. de Melo,

We are pleased to inform you that your manuscript has been judged scientifically suitable for publication and will be formally accepted for publication once it complies with all outstanding technical requirements.

With kind regards,

Francisco C. Santos

Academic Editor

PLOS ONE

Additional Editor Comments (optional):

Reviewers' comments:

Reviewer's Responses to Questions

**Comments to the Author**

1. If the authors have adequately addressed your comments raised in a previous round of review and you feel that this manuscript is now acceptable for publication, you may indicate that here to bypass the “Comments to the Author” section, enter your conflict of interest statement in the “Confidential to Editor” section, and submit your "Accept" recommendation.

Reviewer #1: All comments have been addressed

Reviewer #2: All comments have been addressed

2. Is the manuscript technically sound, and do the data support the conclusions?

Reviewer #1: Yes

Reviewer #2: Yes

3. Has the statistical analysis been performed appropriately and rigorously? 

Reviewer #1: Yes

Reviewer #2: Yes

4. Have the authors made all data underlying the findings in their manuscript fully available?

Reviewer #1: Yes

Reviewer #2: Yes

5. Is the manuscript presented in an intelligible fashion and written in standard English?

Reviewer #1: Yes

Reviewer #2: Yes

6. Review Comments to the Author

Reviewer #1: The authors have addressed all my questions and comments. I congratulate them on the work, and believe this is a very interesting paper. I also look forward to further research on this topic.

Reviewer #2: (No Response)

7. PLOS authors have the option to publish the peer review history of their article (what does this mean?). If published, this will include your full peer review and any attached files.

Reviewer #1: No

Reviewer #2: No

---

## [Editor Report · Acceptance letter]

1 Nov 2019

PONE-D-19-20447R1 

Cooperation with Autonomous Machines Through Culture and Emotion 

Dear Dr. de Melo:

I am pleased to inform you that your manuscript has been deemed suitable for publication in PLOS ONE. Congratulations! Your manuscript is now with our production department. 

With kind regards,

on behalf of

Dr. Francisco C. Santos 

Academic Editor

PLOS ONE